# Building Energy Consumption Prediction Using a Deep-Forest-Based DQN Method

Qiming Fu [1,2,†], Ke Li [1,2,†], Jianping Chen [2,3,4,*], Junqi Wang [2] , You Lu [1,2] and Yunzhe Wang [1,2]

1. School of Electronics and Information Engineering, Suzhou University of Science and Technology, Suzhou 215009, China; fqm_1@126.com (Q.F.); 1913041010@post.usts.edu.cn (K.L.); luyou@usts.edu.cn (Y.L.); yunzhe1991@mail.usts.edu.cn (Y.W.)
2. Jiangsu Province Key Laboratory of Intelligent Building Energy Efficiency, Suzhou University of Science and Technology, Suzhou 215009, China; junqi_wang@seu.edu.cn
3. School of Architecture and Urban Planning, Suzhou University of Science and Technology, Suzhou 215009, China
4. Chongqing Industrial Big Data Innovation Center Co., Ltd., Chongqing 400707, China
* Correspondence: alanjpchen@aliyun.com
† These authors contributed equally to this work.

**Abstract:** When deep reinforcement learning (DRL) methods are applied in energy consumption prediction, performance is usually improved at the cost of the increasing computation time. Specifically, the deep deterministic policy gradient (DDPG) method can achieve higher prediction accuracy than deep Q-network (DQN), but it requires more computing resources and computation time. In this paper, we proposed a deep-forest-based DQN (DF–DQN) method, which can obtain higher prediction accuracy than DDPG and take less computation time than DQN. Firstly, the original action space is replaced with the shrunken action space to efficiently find the optimal action. Secondly, deep forest (DF) is introduced to map the shrunken action space to a single sub-action space. This process can determine the specific meaning of each action in the shrunken action space to ensure the convergence of DF–DQN. Thirdly, state class probabilities obtained by DF are employed to construct new states by considering the probabilistic process of shrinking the original action space. The experimental results show that the DF–DQN method with 15 state classes outperforms other methods and takes less computation time than DRL methods. MAE, MAPE, and RMSE are decreased by 5.5%, 7.3%, and 8.9% respectively, and $R^2$ is increased by 0.3% compared to the DDPG method.

**Keywords:** energy consumption prediction; deep forest; deep Q-network; shrunken action space





## 1. Introduction

Global energy consumption increases drastically every year due to economic development and population growth. Building energy consumption is an integral part of the world's total energy consumption, accounting for 20.1% on average [1]. In many countries, this percentage is much higher; for example, it accounts for 21.7% and 38.9% of total energy consumption in China and America, respectively [2,3]. This increasing energy consumption exacerbates global warming and the scarcity of natural resources. Hence, improving building energy efficiency is crucial, as it can slow down global warming and promote the sustainable development.

Energy consumption prediction plays an important role in improving building energy efficiency, since it can facilitate the implementation of many building energy efficiency measures, namely demand response of buildings [4], urban energy planning [5], and fault detection [6]. It can also assist in assessing operation strategies of different systems, such as heating, ventilation, and air conditioning (HVAC) systems [7], and indirect evaporative cooling energy recovery systems [8] to save energy. Therefore, numerous studies have been concerned with energy consumption prediction, and many methods have been introduced to predict energy consumption.

### 1.1. Related Work

1.1.1. Energy Consumption Prediction

According to Ref. [9], all methods for energy consumption prediction can be roughly classified into engineering, statistical, and artificial intelligence methods. Table 1 shows the merits and demerits of these methods.

The engineering methods employ physics principles and thermodynamic formulas to calculate the energy consumption of each component of the building. Relationships between input and output variables are very clear, but these methods require detailed building and environmental information, which is often very difficult to obtain. Some researchers have tried to simplify engineering models to effectively predict energy consumption.

Yao et al. [10] proposed a simple method of formulating load profile (SMLP) to predict the daily breakdown energy demand of appliances, domestic hot water, and space heating. This method predicted energy demand for one season at a time since the average daily consumption for each component varied seasonally. Wang et al. [11] simplified the physical building model to predict cooling load. The parameters of the simplified models of the building envelopes were determined using easily available physical properties based on frequency response characteristic analysis. Moreover, they employed a thermal network of lumped thermal mass to represent a building's internal mass with parameters identified using monitored operation data. However, because they used simplified models, the prediction results may not have been completely accurate [12].

Statistical methods use mathematical formulas to correlate energy consumption data with influencing factors. Ma et al. [13] employed multiple linear regression (MLR) and self-regression methods to construct models based on the analysis of the relevant power energy consumption factors, such as specific population activities and weather conditions. They used the least square method to estimate parameters and predicted monthly power energy consumption for large-scale public buildings. Lam et al. [14] developed a new climatic index based on the principal component analysis (PCA) of three major climatic variables: dry-bulb temperature, wet-bulb temperature, and global solar radiation. Then, regression models were constructed to correlate the simulated daily cooling load with the corresponding daily new index. The calculation processes of these statistical methods are straightforward and fast, but they often cannot handle stochastic occupant behaviors and complex interactions between factors, so they are not flexible and often have poor prediction accuracy [15].

Artificial intelligence methods can learn from historical data, which are usually called data-driven methods, and they usually performance better than other methods [16]. In Ref. [17], all artificial intelligence methods were broadly divided into two categories: traditional artificial intelligence methods and deep learning methods. However, in practice, decision tree (DT), support vector machine (SVM), artificial neural network (ANN), and many other traditional machine learning methods can be regarded as traditional artificial intelligence methods. Azadeh et al. [18] proposed a method based on ANN and analysis of variance (ANOVA), which was used to predict annual electricity consumption. The method was more effective than the conventional regression model. Hou et al. [19] employed SVM to predict the cooling load of HVAC system, and the results indicated that the SVM method was better than auto-regressive integrated moving average (ARIMA) methods. In Ref. [20], DT, stepwise regression, and ANN methods were employed to predict electricity energy consumption in Hong Kong. The prediction results indicated that DT and ANN methods performed slightly better in the summer and winter phases, respectively. However, these traditional artificial intelligence methods adopt shallow structures for modeling, limiting models' prediction accuracy.

Deep learning (DL) methods may not always reflect physical behaviors, but they can learn more abstract features from raw inputs to construct better models [21]. Cai et al. [22] used recurrent neural network (RNN) and convolutional neural network (CNN) to forecast time-series building-level load in recursive and direct multi-step manners. The experimental results showed that the gated 24-h CNN method could improve prediction

accuracy by 22.6%, compared with the seasonal auto-regressive integrated moving average with exogenous inputs (ARIMAX). Ozcan et al. [23] proposed dual-stage attention-based recurrent neural networks to predict electric load consumption. The method used the encoder and decoder for feature extraction as well as the attention mechanism. The experimental results indicated that the proposed method outperforms other methods.

Deep reinforcement learning (DRL) methods are another category of artificial intelligence methods that cannot be neglected. DRL methods combine the perception of DL with decision-making of reinforcement learning (RL) and have achieved many substantial results in many fields, such as games [24], robotics [25], and autonomous driving [26]. In the building field, DRL methods are mainly used to find the optimal control in HVAC systems [27,28]. Many researchers have also used DRL methods for energy consumption prediction and achieved satisfactory results. For instance, Liu et al. [29] explored the performance of DRL methods for energy consumption prediction, and the results showed that the deep deterministic policy gradient (DDPG) method achieved the highest prediction accuracy in single-step-ahead prediction. However, the potential of DRL methods has not been fully realized. One limitation of current works is that many researchers only focus on the DDPG method, but ignore the classical deep Q-network (DQN) method.

**Table 1.** Summary of merits and demerits of the prediction methods.

| Method | | Merits | Demerits |
|---|---|---|---|
| Engineering [10,11] | | Relationships between input and output variables are very clear | Detailed building information is required |
| Statistical [13,14] | | Straightforward and fast | Not flexible |
| Artificial intelligence | Traditional machine learning [18–20] | Learn from historical data | Adopt shallow structures for modeling |
| | Deep learning [21,22,29] | Extract more abstract features from raw inputs | May not always reflect the physical behaviors |

### 1.1.2. Predictive Control

Predictive control is a multivariable control strategy based on prediction, and its aim is to minimize the cost function [30]. The predictive control strategy can reduce energy consumption and improve energy efficiency in the building field. Shan et al. [31] utilized chiller inlet guide vane openings as an indicator of chiller efficiency and cooling load to develop a robust chiller sequence control strategy. In the strategy, the opening or closing of a chiller was based on the measured cooling load and the predicted maximum cooling capacity, which was obtained by the MLR method. The experiment showed that the strategy could save 3% energy in a tested building compared with a typical strategy. In Ref. [32], predictive control was applied to space heating buildings. The heating demand of buildings was predicted using EnergyPlus software. Then, the predictive control strategy selected the appropriate operation schedule to minimize the electricity cost while meeting the heating demand of buildings. Finally, different buildings achieved cost savings of around 12–57% through a 7-day simulation. Imran et al. [33] proposed an IoT task management mechanism based on predictive optimization to minimize the energy cost and maximize thermal comfort. In this mechanism, the predictive optimization was based on the hybrid of prediction and optimization and used to optimize and control energy consumption. It was shown that predictive optimization-based energy management outperformed standalone prediction and optimization mechanisms in smart residential buildings.

In predictive control, methods of prediction and control are different. Prediction methods are usually employed to solve this regression problem, and control methods are used to find the optimal solution. DRL can also be used in predictive control as a model-free control method. For instance, Qin et al. [34] proposed a multi-discipline predictive

intelligent control method for maintaining thermal comfort in an indoor environment. SVR was employed to construct an environmental prediction model, then the DRL method was applied to train an intelligent agent for intelligent control in the indoor environment. The results showed that the predictive control method could ensure thermal comfort and air quality in the indoor environment while minimizing energy consumption. Fu et al. [35] established a thermal dynamics model to predict the future trend of HVAC systems. Twin delayed deep deterministic policy gradient algorithm and model predictive control (TD3–MPC) were proposed to pre-adjust building temperatures at off-peak times. It was shown that TD3–MPC reduced energy consumption cost by 16% and thermal comfort RMSE by 0.4.

DRL methods can also be used to solve prediction problems. Some researchers have noted the power of the DDPG method, and use this method to predict HVAC system energy consumption [36]. However, the classical DQN method is often neglected and is rarely used for building energy consumption prediction.

*1.2. The Purpose and Organization of This Paper*

To date, the DDPG method has been investigated and developed more than other methods since it can process continuous action space problems; the DQN method with discrete action space is usually neglected. However, it cannot be ignored that the DDPG method often needs more computing resources and computation time to achieve high prediction accuracy. In contrast, the DQN method may not achieve such high prediction accuracy, but it can take less computation time than the DDPG method.

To obtain a higher prediction accuracy than the DDPG method and take less computation time than the DQN method, a deep-forest-based DQN (DF–DQN) method is proposed. The main contributions of this paper are as follows:

(1) The shrunken action space is proposed to replace the original action space, then DF–DQN method can quickly obtain the optimal action in the shrunken action space.

(2) State classes obtained by deep forest (DF) are used to determine the specific meaning of each action in the shrunken action space, and they can map the shrunken action space to a single sub-action space. Hence, the convergence of the DF–DQN method can be ensured.

(3) New states, composed of state class probabilities and historical energy consumption data, are constructed to improve the robustness of the DF–DQN method.

The remainder of this paper is structured as follows. In Section 2, theories of DQN and DF methods are simply described. Section 3 presents the overall framework of the DF–DQN method for energy consumption prediction. Then, the procedure of data pre-processing and MDP modeling are described in detail. Section 4 depicts experimental settings and adopted metrics of all methods, and experimental results are compared and analyzed. Some conclusions are given in Section 5.

## 2. Related Theories

*2.1. Deep Reinforcement Learning*

2.1.1. Reinforcement Learning

RL is an essential branch of machine learning; its final goal is to maximize the accumulative discount reward $R_t$ [37], as shown in Equation (1):

$$R_t = \sum_{k=0}^{\infty} \gamma^k r_{t+k+1} \tag{1}$$

where $\gamma$ is a discount factor and $k$ represents different time steps. $r_{t+k+1}$ represents the immediate reward in different time steps. Generally, RL problems can be modeled as a Markov decision process (MDP) to be solved. A MDP is a five-tuple $(S, A, P, R, \gamma)$, where $S$ is a set of states, $A$ is a set of actions, $P$ is a transition function, and $R$ is a reward function. In the process of an agent interacting with the environment, the agent receives a state

$s_t$ and executes an action $a_t$ at time step $t$. Notably, the action $a_t$ is selected by policy $\pi$, which represents a mapping from the state space $S$ to the action space $A$. Then, the agent is transferred to the next state $s_{t+1}$, which is determined from the probability of state transition $P(s_{t+1}|s_t, a_t)$. Simultaneously, the immediate reward $r_{t+1}$ is obtained by the environment.

In RL methods, the action value function $Q$ represents the expectation of accumulative discount reward starting from state $s$ and taking action $a$:

$$Q_\pi(s,a) = E_\pi\left[\sum_{k=0}^{\infty} \gamma^k R_{t+k+1}|s_t = s, a_t = a\right] \tag{2}$$

Policy $\pi$ can be evaluated and improved by the action value function and optimal action value function, which can be denoted as:

$$Q_*(s,a) = \max_\pi Q_\pi(s,a) = E\left[R_{t+1} + \gamma\max_{a'} Q_*(s_{t+1}, a')\middle| s_t = s, a_t = a\right] \tag{3}$$

Finally, the optimal policy $\pi_*$ is obtained, and the final goal $R_t$ can be achieved by the optimal policy.

### 2.1.2. Deep Q-Network

Traditional RL methods, such as Q-learning and SARSA [38,39], can only tackle tasks with state spaces that are small and discrete. Recent methods have diverged from these restrictions by employing the deep neural network to approximate the action value function. However, these methods usually are not stable as they combine RL methods with function approximation, such as linear function or deep neural network [40]. This problem has recently been overcome by DQN with two specific techniques.

Firstly, the mechanism of experience replay is adopted to remove strong correlations between successive inputs, which means that experience tuples are stored in the replay memory and sampled randomly to train an agent. Here, experience tuples are generated by the interaction between the agent and the environment.

Secondly, to reduce correlations with targets, a separate network, namely the target Q-network, is constructed to generate targets and update the Q-network. Specifically, every $J$ steps, the target Q-network is obtained by cloning the Q-network to calculate targets for the following $J$ steps. The loss function at iteration $i$ can be formulated by using two networks:

$$L(\theta_i) = E[(r + \gamma\max_{a'} Q(s', a'|\theta_i^-) - Q(s, a|\theta_i))^2] \tag{4}$$

where $(s, a, r, s')$ is an experience tuple sampled in the replay memory, and $a'$ is the selected action in state $s'$. $\theta_i^-$ and $\theta_i$ represent the parameters of the target Q-network and Q-network, respectively.

### 2.2. Deep Forest

DF is a novel decision tree ensemble method that can be applied for classification tasks [41]. Two techniques, namely multi-grained scanning and cascade forest structure, improve the performance of DF.

In the procedure of multi-grained scanning, all training samples are first transformed into instances using a sliding window. Then, all instances are employed to train a certain number of completely random tree forests and random forests, as well as the class vectors are generated. Finally, the transformed feature vectors are obtained by concatenating these class vectors.

The cascade forest structure is used to enhance the representational learning ability of DF. Each level receives feature information processed by its preceding level and outputs its processing result to the next level. The transformed feature vectors, which are the outputs

of multi-grained scanning, are used to train the first level of the cascade forest. The final results are obtained at the last level and expressed as the maximum aggregated value.

## 3. DF–DQN Method for Energy Consumption Prediction

### 3.1. Overall Framework

Figure 1 depicts the overall framework of DF–DQN method for energy consumption prediction. The energy consumption data is divided into a training set and a test set according to the date. Then, the method, which can identify local outliers, is executed to detect outliers in the training set, and outliers are replaced considering date attribute and time factor. Feature extraction is then conducted to select $h$ historical data as features, as well as samples and their corresponding labels can be constructed by these features. In addition, the features of each sample need to be normalized before they are transmitted into DF and DQN, which can enhance prediction accuracy.

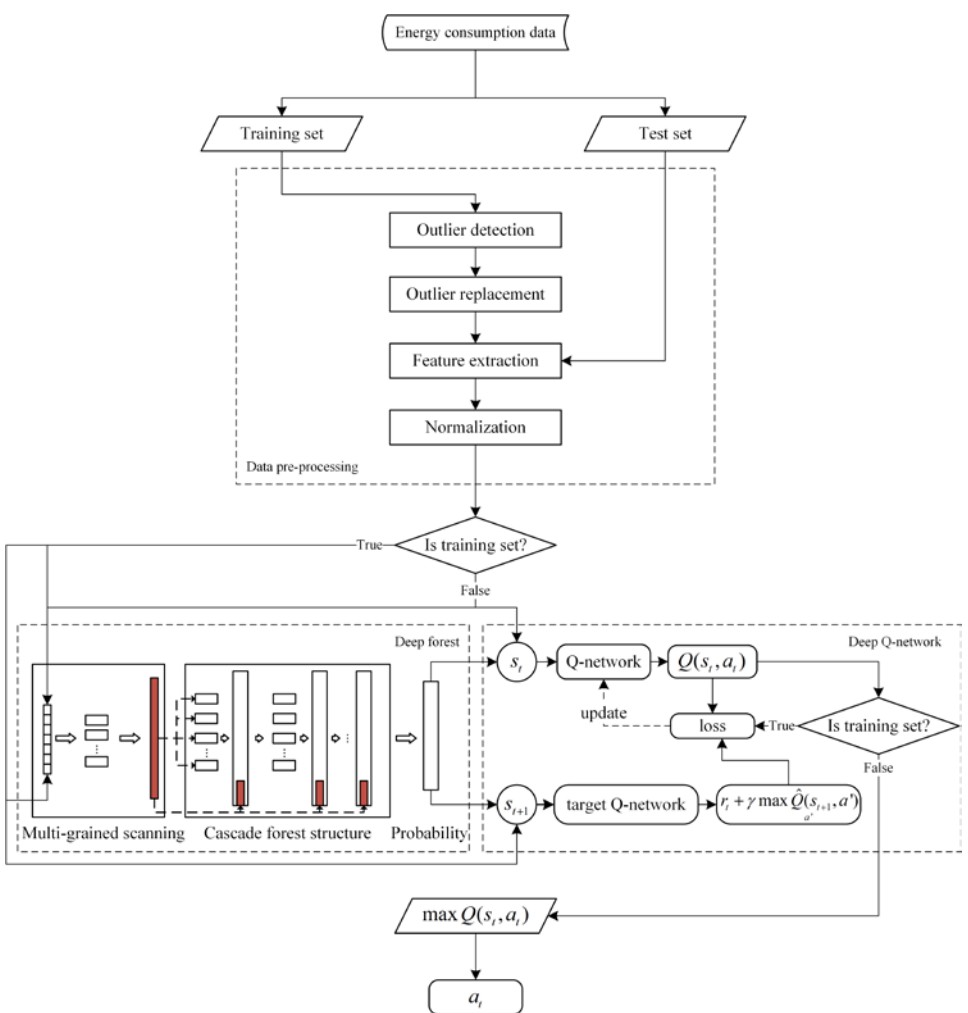

**Figure 1.** Overall framework of the DF–DQN method.

In the training process, a DF classifier is trained, firstly, by samples and labels, which are generated by the training set. Once the classifier training is complete, the normalized samples can be passed into a DF classifier as raw feature vectors. The transformed feature vectors are obtained in the procedure of multi-grained scanning. Then, the cascade forest structure takes the transformed feature vectors as inputs and outputs the probabilities of each class. The state at time step $t$ (i.e., $s_t$) can be constructed, which is composed of the normalized historical data and the corresponding probabilities of each class. The Q-network takes $s_t$ as input to calculate $Q$ values for all actions, and an action with probability $1 - \varepsilon$

that has the maximum $Q$ value or a random action with probability $\varepsilon$ should be selected. The action selected at time step $t$ (i.e., $a_t$) is the predicted energy consumption. Hence, the immediate reward can be obtained by the predicted energy consumption and the actual energy consumption at time step t. Similarly, the target Q-network can take the state at time step $t + 1$ (i.e., $s_{t+1}$) as input to calculate target $Q$ values for all actions. Finally, loss can be calculated by $Q$ values and target $Q$ values, and employed to update the Q-network.

In the test process, when the DF–DQN method receives $h$ normalized historical data, the probabilities of each class are first obtained by DF. Then, a new state is constructed, and $Q$ values for all actions are calculated. The action with the highest $Q$ value is selected as the predicted energy consumption.

*3.2. Data Pre-Processing*

In this study, the data set concerned was the energy consumption data in an office building in Shanghai. The energy consumption data was collected every hour from 1 January 2015 to 31 December 2016. There were a total of 17,520 observation samples as the data from 29 February 2016 was not collected.

Since there may have been mixed use of electric meters, outliers were generated with high probabilities. We first detected outliers to improve the accuracy of energy consumption prediction. The local outlier factor (LOF) method is a density-based unsupervised method for identifying local outliers [42]. To find possible outliers, it can calculate local density deviation (i.e., LOF value) for each sample to their neighbors. If LOF values are high, samples have high probabilities of being treated as outliers. Similarly, samples with lower LOF values are more likely to be considered as normal data. Therefore, the LOF method can detect abnormal energy consumption data in the training set.

Outliers cannot be simply discarded after they are detected. The energy consumption data were collected at intervals of 1 h and had time-series periodicity. The absence of outliers would make the data series less accurate and make feature extraction and method execution more difficult. Therefore, outliers were replaced, which was more conducive to energy consumption prediction.

The replacement of outliers requires consideration of the time factor. In addition, the impact of holidays is not negligible for office buildings. These factors should be considered when replacing outliers. If the energy consumption data of workday is an outlier, it can be replaced by the average value, which is calculated by the sum of normal energy consumption data at the same time on the previous and following workday. The method used to replace holiday outliers is the same. This process can be formulated as below:

$$AE(d,t) = \begin{cases} NE(d-i,t) & condition1 \\ \frac{NE(d-i,t)+NE(d+j,t)}{2} & condition2 \\ NE(d+j,t) & condition3 \end{cases} \tag{5}$$

$$condition1. d-i \geq p, d+j > q, W(d-i) = W(d)$$
$$condition2. d-i \geq p, d+j \leq q, W(d-i) = W(d) = W(d+j)$$
$$condition3. d-i < p, d+j \leq q, W(d) = W(d+j)$$

where $i, j \in N$, $AE$, and $NE$ denote the abnormal and normal energy consumption data. The date and time of energy consumption data are represented by $d$ and $t$ respectively. $W(d)$ can determine the date attribute of $d$ (workday or holiday), and $p$ and $q$ represent the start and end date in the training set.

In this study, the date range of the training set was from 1 January 2015 to 31 October 2016. If one of the two normal energy consumption dates is outside the range, the outlier can be directly replaced with other normal energy consumption data. The dates of two normal energy consumption data used to replace outliers usually do not simultaneously fall outside the date range in the training set. The reason for this is that the date span is large. It should be mentioned that the above process of detecting and replacing outliers

can only be performed in the training set, and the test data, which was unknown, cannot be used.

Feature extraction is the next step, selecting a certain number of historical energy consumption data as features. We set the number to $h$. If the energy consumption at time step $t$ (denoted as $E_t$) needs to be predicted, $h$ historical data from $t - h$ to $t - 1$ can be selected as features, which can be denoted as $(E_{t-h}, \ldots, E_{t-1})$. In other words, $(E_{t-h}, \ldots, E_{t-1})$ is used to predict the actual energy consumption at time step t. $(E_{t-h}, \ldots, E_{t-1})$ can be regarded as a sample, and $E_t$ is the corresponding label. Therefore, when the total number of training data is $M$, $M - h$ samples and labels can be constructed.

In order to improve the accuracy of energy consumption prediction, each feature of the samples should be normalized as below:

$$\widetilde{X}_i^{(j)} = \frac{X_i^{(j)} - \mu^{(j)}}{\sigma^{(j)}} \tag{6}$$

where $X_i^{(j)}$ and $\widetilde{X}_i^{(j)}$ denote the previous and normalized value of $j$-th feature of the $i$-th sample, and $\mu^{(j)}$ and $\sigma^{(j)}$ denote the mean and standard deviation of the $j$-th feature, respectively.

### 3.3. MDP Modeling

When the DF–DQN method is employed to predict energy consumption, the prediction problem should be transformed into a control problem. This means that the process of energy consumption prediction should be modeled as an MDP, and the state, action and reward function should be defined.

The MDP constructed by the DF–DQN method improves settings in DQN method for energy consumption prediction. When the DQN method predicts energy consumption, all states are previous normalized samples. Specifically, $h$ normalized historical energy consumption data, which is denoted as $\left( \widetilde{E}_{t-h}, \widetilde{E}_{t-h-1}, \ldots, \widetilde{E}_{t-1} \right)$, compose the state at time step $t$ (i.e., $s_t$). In terms of the setting of actions, the range of historical energy consumption data determines the number of actions and action values, which is the predicted energy consumption. If the range of historical data is $[x, z]$, and the step size is $g$, the action selected at time step $t$ (i.e., $a_t$) can be selected from $\{x, x + g, x + 2g, \ldots, z\}$, which represents the original action space, and the total number of actions is $(z - x)/g + 1$. By contrast, the DF–DQN method shrinks the original action space and introduces the DF classifier for energy consumption prediction.

#### 3.3.1. Shrunken Action Space

This section describes the procedure of shrinking the original action space. Assuming that the historical data range is {10,59} and the step size is 1, the original action space X is generated as depicted in Figure 2. It should be mentioned that the action value is equal to the predicted energy consumption in the action space X. For example, the value of first action is 10, which means that the predicted energy consumption is 10. In practice, all action values in the action space X can be converted into other forms. Figure 2 illustrates this process, where action values of 10, . . . , 59 are transformed into 10 + 0, . . . , 50 + 9. Therefore, the original action space X can be replaced with the action space Y, and the action space size is not changed.

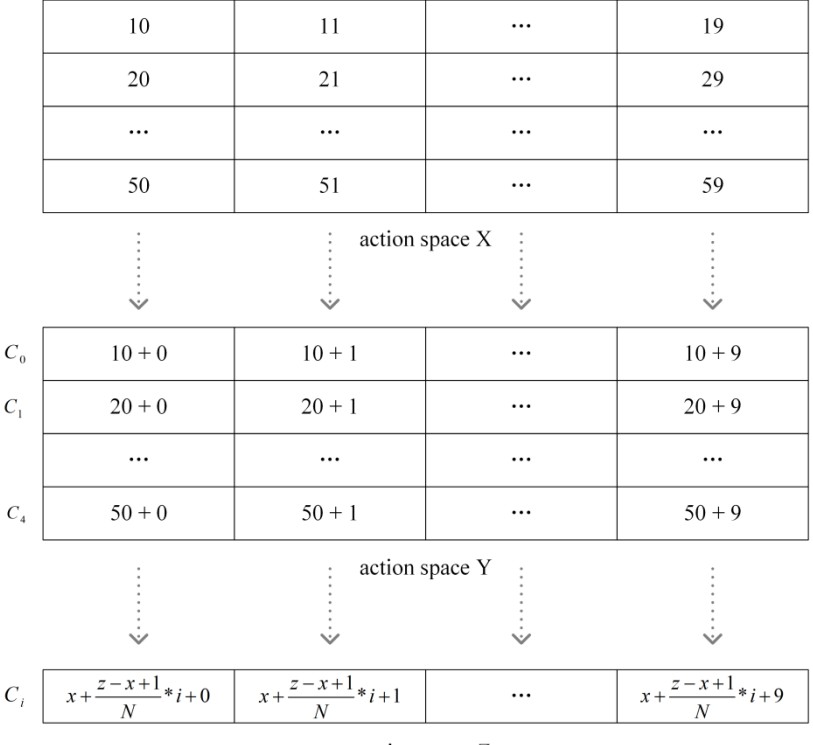

**Figure 2.** An example of shrinking action space.

The action space Y can be equally divided into a certain number of sub-action spaces, and all states corresponding to each sub-action space can be classified into one class. Therefore, we assume that each row of actions in the action space Y composes a sub-action space, and its corresponding states are classified into one class. Finally, five sub-action spaces are generated, and all states that compose the state space can be classified into five classes (i.e., $C_0, C_1, C_2, C_3, C_4$). As shown in Figure 2, $C_0$ represents that all states that correspond to actions in the first sub-action space are classified into class 0. Similarly, the meaning of $C_1$, $C_2$, $C_3$, and $C_4$ can be obtained.

After the state classes are determined, the actions of the same sequence in all sub-action spaces can be represented uniformly. This is because the correlation of these actions can be established by using state classes. The exact formula is denoted below:

$$x + \frac{z - x + 1}{N} \times i + j \qquad i = 0, 1, 2, 3, 4. \qquad j = 1, \ldots, 9 \tag{7}$$

where $x$ and $z$ are equal to 10 and 59, which are the lower and upper bounds of the range of historical data. $N$ and $i$ denote the number of state classes and the $i$-th state class, respectively, and $j$ represents the $j$-th action in the shrunken action space. The final result is shown at the bottom of Figure 2. Fifty actions in the action space Y are replaced with ten actions in the action space Z. At this point, Z can be regarded as the shrunken action space. Note that the step size is set to 1, and the number of actions in each sub-action space is 10 in the above example. In a more general sense, the step size is $g$, and the number of actions in each sub-action space is $n$. The actions of same sequence in all sub-action spaces can be represented as below:

$$x + \frac{\frac{z-x}{g} + 1}{N} \times i + j \qquad i = 0, 1, \ldots, N - 1. \qquad j = 0, g, 2g, \ldots, n \tag{8}$$

The result of applying this formula is that $N \times n$ actions in the original action space are replaced with $n$ actions in the shrunken action space.

### 3.3.2. DF Classifier

Unlike the DQN method, the DF–DQN method needs a trained DF classifier. A DF classifier is chosen because it is not sensitive to hyperparameters and is easy to train. In Ref. [41], it was shown that DF could achieve excellent performance using almost the same settings of hyper-parameters when it is applied to different data across different domains. A DF classifier is introduced for two purposes. First, it can map the shrunken action space to a single sub-action space. Second, the state class probabilities obtained by the DF classifier are employed to construct new states.

The training of the DF classifier is based on the shrunken action space. In the process of shrinking the original action space, each sub-action space is generated by the range of energy consumption data. Further, different data ranges indicate different classes. As shown in Figure 3, {10–19} can be regarded as one sub-action space, and the data range {10–19} can represent the class $C_0$. Assuming that a sample obtained by feature extraction is $(E_{t-h}, E_{t-h+1}, \ldots, E_{t-1})$ and its corresponding label is $E_t$, if $E_t$ is within the range, the extra class label of $(E_{t-h}, E_{t-h+1}, \ldots, E_{t-1})$ is $C_0$. Therefore, all samples obtained by feature extraction have additional class labels since their original labels must be within the energy consumption range of a sub-action space. These samples and class labels can compose the training set to train a DF classifier.

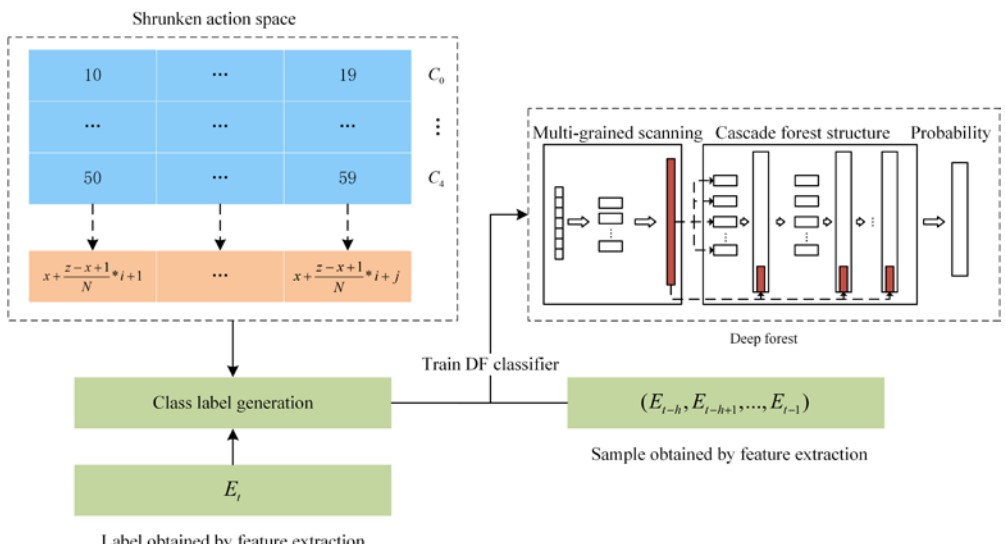

**Figure 3.** The training process of the DF classifier.

In the DF–DQN method, the input of the trained DF classifier is the original state composed of historical energy consumption data, and outputs are state class probabilities. These probabilities serve two purposes. Firstly, they can determine a single sub-action space. Secondly, they can be deployed to construct new states.

In the shrunken action space, each action has multiple meanings. For instance, the first action in the shrunken action space can represent the first action of each sub-action space. Similarly, in the $Q$ neural network, one action has multiple meanings, which means that one neuron is used to represent multiple actions. The result is that the $Q$ network cannot determine the specific meaning of the neuron at each time step, so it cannot converge effectively. State class probabilities, which are obtained by the DF classifier, can map the shrunken action space to a single sub-action space, then determine the specific meaning of each action. Assuming that the total number of classes is five, and the state class probabilities are (0.7, 0.05, 0.1, 0.1, 0.05), the corresponding state can be regarded as the first class. The shrunken action space is then equal to the first sub-action space, and it can be denoted as {10, 11, . . . , 19}. This process is shown in Figure 4.

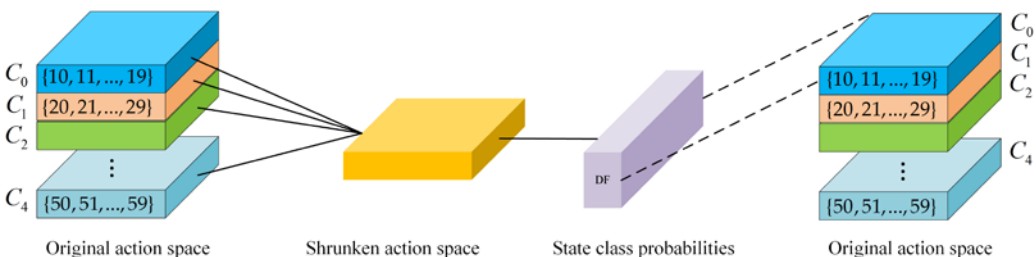

**Figure 4.** Mapping the shrunken action space to a single sub-action space.

Further, we constructed new states composed of the normalized historical data and its corresponding state class probabilities. This is because mapping the shrunken action space to a single sub-action space is a probability process. The Q-network should consider the probability factor when calculating the $Q$ value, so new states are constructed to replace the original state. Moreover, new states can also be seen as integration. The performance of the DF classifier is integrated with the decision-making ability of DQN to enhance the prediction accuracy for energy consumption prediction.

Finally, the state of the DF–DQN method at time step $t$ (i.e., $s_t$) is composed of normalized historical energy consumption data and corresponding state class probabilities, which is represented by a vector $(P_0, P_1, \ldots, P_{N-1}, \widetilde{E}_{t-h}, \widetilde{E}_{t-h-1}, \ldots, \widetilde{E}_{t-1})$. Here, $P_i$ denotes the probability that $(\widetilde{E}_{t-h}, \widetilde{E}_{t-h-1}, \ldots, \widetilde{E}_{t-1})$ is determined to $i$-th class and $(\widetilde{E}_{t-h}, \widetilde{E}_{t-h-1}, \ldots, \widetilde{E}_{t-1})$ represents $h$ normalized historical data. The action selected at time step $t$ is $a_t$, which denotes that the predicted energy consumption is $a_t$ at time step $t$. The immediate reward function can then be set as follows:

$$r_{t+1} = -|E_t - a_t| \tag{9}$$

where $E_t$ represents the actual energy consumption at time step $t$. It should be mentioned that the closer the reward is to zero, the higher the prediction accuracy of the DF–DQN method.

### 3.4. DF–DQN Method

Once the problem of energy consumption prediction is modeled as an MDP, the DF–DQN method can be executed. Algorithm 1 depicts the main training process of the DF–DQN method for energy consumption prediction.

---

**Algorithm 1** DF–DQN method for energy consumption prediction

---

(1)　 Initialize state classes $N$
(2)　 Initialize replay memory $D$
(3)　 Initialize action-value function $Q$ with random weights $\theta$
(4)　 Initialize target action-value function $\hat{Q}$ with weights $\theta^- = \theta$
(5)　 Split the data set
(6)　 Detect and replace outliers in the training set
(7)　 Extract features to construct samples and labels
(8)　 Train the deep forest classifier
(9)　 **Repeat** (for each episode)
(10) Randomly select a sample
(11) Use DF classifier to obtain the possibility of each class
(12) Construct initial state (denoted as $s_t$)
(13) **Repeat** (for each step)
(14) Select a random action $a_t$ with probability $\varepsilon$
(15) otherwise choose $a_t = \arg\max Q(s_t, a; \theta)$
(16) Execute action $a_t$ and receive immediate reward $r_t$
(17) Construct state $s_{t+1}$
(18) Store transition $(s_t, a_t, r_t, s_{t+1})$ in $D$
(19) Sample a mini-batch $\left(s_j, a_j, r_j, s_{j+1}\right)$ from $D$
(20) Set $y_j = \begin{cases} r_j & \text{if episode terminates at step } j+1 \\ r_j + \gamma\max\limits_{a'}\hat{Q}\left(s_{j+1}, a' \middle| \theta^-\right) & \text{otherwise} \end{cases}$
(21) Update $Q$ function using $\left(y_j - Q\left(s_j, a_j; \theta\right)\right)^2$
(22) Every $J$ steps reset $\hat{Q} = Q$
(23) $s_t \leftarrow s_{t+1}$
(24) **Until** terminal state or maximum number of steps is reached
(25) **Until** maximum number of episodes is reached

---

## 4. Case Study

The MLR, SVR, DT, DQN, DF–DQN, and DDPG methods were used for energy consumption prediction. Sections 4.1 and 4.2 describe the experimental settings of all methods and four evaluation metrics of prediction accuracy. In Section 4.3, all methods are compared and analyzed from three perspectives, namely prediction accuracy, convergence rate, and computation time.

### 4.1. Experimental Settings

In the process of feature extraction, historical energy consumption data for the last 24 h were selected as features to predict future energy consumption. Hence, there were 24 neurons in the input layer of the DQN and DDPG methods. In contrast, the inputs of the DF–DQN method were composed of historical data and state class probabilities, so the number of neurons in the input layer was $24 + N$ (i.e., the number of state classes). In addition, the range of energy consumption was (129, 1063) in the training set, which means that all methods were performed in the continuous action space. Because the DQN and DF–DQN methods can only process discrete problems, the continuous action space was first converted into a discrete action space. We set the step size to 1 and the range of (129, 1063) was converted into 935 discrete points. The number of actions in the DQN method and the DF–DQN method were 935 and $935/N$, respectively.

The hardware platform and package version used in the study are described in Tables 2 and 3, respectively. In addition, the hyper-parameters of all methods are summarized in Table 4. The DQN, DF–DQN, and DDPG methods used two hidden layers, and the number of neurons in each hidden layer was 32. In regard to the output layer, the number of neurons in the DQN and DF–DQN methods were consistent with the number of actions, while the DDPG method directly outputted the predicted energy consumption so that

the number of neurons was 1. Further, the hyper-parameters of other methods obtained through extensive numerical experiments are also listed in Table 4.

**Table 2.** Hardware platform.

| Hardware Platform | Configuration |
|---|---|
| Operating system | Windows 10 |
| RAM | 8 GB |
| CPU | Intel Core i5-9500 |
| Programing language | Python |
| Programing software | PyCharm |

**Table 3.** Package version.

| Package | Version |
|---|---|
| TensorFlow | 2.2.0 |
| TensorLayer | 2.2.3 |
| NumPy | 1.19.4 |
| pandas | 1.1.5 |
| DeepForest | 0.1.4 |

**Table 4.** Hyper-parameters of all methods.

| Method | Parameters | Results |
|---|---|---|
| MLR | / | / |
| SVR | Kernel function | Linear |
| DT | Evaluation function | Mean squared error |
| | Maximum depth of the tree | 16 |
| DQN | Neurons | 24,32,32,935 |
| | Activation function | ReLu |
| | Learning rate | 0.01 |
| DF–DQN | Neurons | 24+$N$,32,32,935/$N$ |
| | Activation function | ReLu |
| | Learning rate | 0.01 |
| DDPG | Neurons (actor) | 24,32,32,1 |
| | Activation function (actor) | ReLu |
| | Learning rate (actor) | 0.001 |
| | Neurons (critic) | 24,32,32,1 |
| | Activation function (critic) | ReLu |
| | Learning rate (critic) | 0.001 |

*4.2. Evaluation Metrics*

In order to compare the prediction accuracy of all methods, four evaluation metrics were adopted in this study, namely mean absolute error (MAE), mean absolute percentage error (MAPE), root mean square error (RMSE), and coefficient of determination ($R^2$). These evaluation metrics can be denoted as:

$$\text{MAE} = \frac{1}{m} \sum_{i=1}^{m} \left| y_i - y_i' \right| \tag{10}$$

$$\text{MAPE} = \frac{1}{m} \sum_{i=1}^{m} \left| \frac{y_i - y_i'}{y_i} \right| \tag{11}$$

$$\text{RMSE} = \sqrt{\frac{1}{m} \sum_{i=1}^{m} (y_i - y_i')^2} \tag{12}$$

$$R^2 = 1 - \frac{\sum_{i=1}^{m}(y_i - y_i')^2}{\sum_{i=1}^{m}(y_i - \overline{y})^2} \tag{13}$$

where $m$ denotes the total number of samples, $y_i$ and $y_i'$ represent the predicted value and actual value of the $i$-th sample, respectively. $\overline{y}$ is the average of actual value.

### 4.3. Results and Analyses

In this section, each method was trained ten times under the settings of the hyper-parameters presented in Table 4, and all experimental results were obtained as averages.

#### 4.3.1. Prediction Accuracy

Figure 5 illustrates predicted results of the DF–DQN method with different state classes, where the horizontal axis reflects the predicted energy consumption and the vertical axis represents the actual energy consumption. In each sub-figure, the solid blue line indicates that the predicted energy consumption is equivalent to the actual energy consumption, and the blue dashed line denotes the 20% error. The shaded area represents that the predicted value differs less than 20% from the actual value. Therefore, the number of predicted points contained in the shaded area can reflect the prediction accuracy. The DF–DQN method with 15 and 19 state classes outperformed others, and the DF–DQN method with three state classes was the least effective. The predicted points show the trend of classification since the states were classified using the DF–DQN method. Specifically, the classification trend was noticeable in in the DF–DQN method with 5, 7, and 11 state classes.

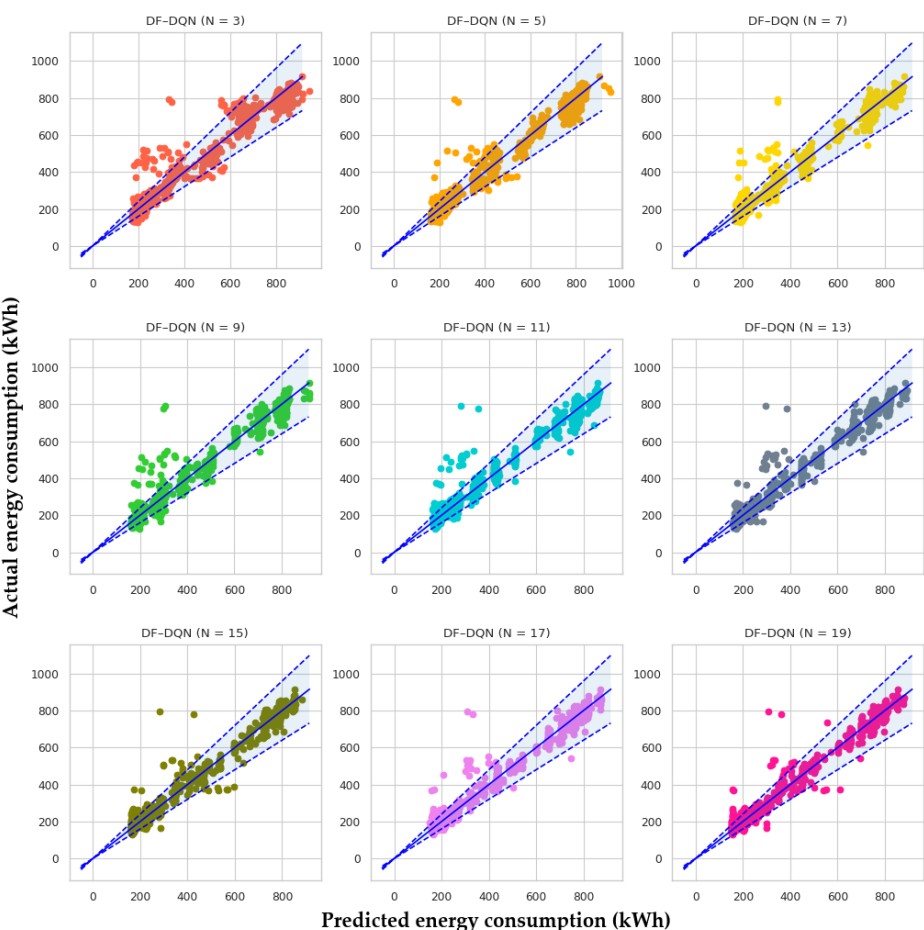

**Figure 5.** Predicted results of the DF–DQN method with different state classes.

Table 5 describes the prediction accuracy of the DF–DQN method with different state classes under four evaluation metrics, and the significant values are indicated in bold.

Notably, the DF–DQN method with 15 state classes had the lowest MAE, MAPE, and RMSE values, and the highest $R^2$ value. Therefore, the prediction accuracy of the DF–DQN method with 15 state classes was the highest. Table 5 also shows that the prediction accuracy increased roughly as state classes increased. However, some unexpected results were observed due to the classification accuracy and the influence of random factors in the training process. For example, the prediction errors of the DF–DQN method with two, four, and five state classes were lower than that of the DF–DQN method with six and seven state classes. Nonetheless, the trend that the increasing number of state classes improved the prediction accuracy was not affected. It also should be mentioned that excessive state classes decreased the prediction accuracy of the DF–DQN method due to the low classification accuracy. Therefore, the optimal number of state classes requires extensive experiments to obtain. In this experiment, the DF–DQN method with 15 state classes outperformed other numbers of state classes.

**Table 5.** Prediction accuracy of the DF–DQN method with different state classes.

| N | Number of Actions | Accuracy of Classification | MAE | MAPE | RMSE | $R^2$ |
|---|---|---|---|---|---|---|
| 2 | 468 | 99.392% | 23.333 | 7.960% | 36.774 | 0.978 |
| 3 | 312 | 94.706% | 29.630 | 9.664% | 48.750 | 0.961 |
| 4 | 234 | 96.168% | 22.950 | 7.828% | 39.141 | 0.974 |
| 5 | 187 | 95.178% | 23.357 | 7.866% | 38.686 | 0.976 |
| 6 | 156 | 92.739% | 27.439 | 9.476% | 44.295 | 0.968 |
| 7 | 134 | 89.583% | 27.512 | 9.655% | 41.153 | 0.971 |
| 8 | 117 | 89.098% | 23.810 | 8.074% | 39.536 | 0.974 |
| 9 | 104 | 88.327% | 23.152 | 7.886% | 40.831 | 0.973 |
| 10 | 94 | 84.139% | 24.456 | 8.370% | 42.201 | 0.970 |
| 11 | 85 | 83.907% | 23.643 | 8.246% | 40.044 | 0.974 |
| 12 | 78 | 83.586% | 22.254 | 7.845% | 37.480 | 0.976 |
| 13 | 72 | 80.307% | 21.921 | 7.379% | 36.248 | 0.978 |
| 14 | 67 | 77.548% | 20.912 | 7.231% | 34.936 | 0.980 |
| 15 | 63 | 76.462% | **20.432** | **7.021%** | **34.057** | **0.981** |
| 16 | 59 | 73.005% | 20.975 | 7.390% | 34.331 | 0.980 |
| 17 | 55 | 71.633% | 20.971 | 7.545% | 35.410 | 0.980 |
| 18 | 52 | 69.037% | 20.590 | 7.315% | 35.442 | 0.979 |
| 19 | 50 | 66.714% | 20.596 | 7.367% | 34.408 | 0.980 |
| 20 | 47 | 64.740% | 20.623 | 7.272% | 35.198 | 0.980 |

In addition, results using the DQN, DDPG, MLR, SVR, and DT methods were compared with those using the DF–DQN method. As shown in Figure 6, the DF–DQN method with 15 state classes and the DDPG method outperformed other methods. In order to further analyze the prediction accuracy of all methods, predicted results of a certain period were selected for display. Note that the energy consumption on workdays and holidays should only be compared with other results of their type, as the results from each category are quite different. Figure 7 depicts predicted results of all methods on workdays, where the horizontal axis reflects the time and the vertical axis represents the energy consumption. In each sub-figure, the blue line represents actual energy consumption, and the line of another color denotes the predicted energy consumption. It can be seen that all methods captured the energy consumption trend, and only the MLR method showed slight fluctuation. In contrast, the predicted results on holidays were inaccurate, as seen in Figure 8. The DQN, MLR, and SVR methods showed fluctuation, and the DT method had a noticeable error point. The DDPG method and the DF–DQN method with 15 state classes also captured part of the trend of actual energy consumption. A possible explanation for this might be that all methods were unable to learn the features of holiday energy consumption when only historical data were used as input.

Table 6 describes the prediction accuracy of all methods in detail. Notably, the DDPG method had advantages in energy consumption prediction, and its prediction accuracy was higher than the MLR, SVR, DT, and DQN methods. However, for the DF–DQN

method with 15 state classes, MAE, MAPE, and RMSE decreased by 5.5%, 7.3%, and 8.9% respectively, and $R^2$ increased by 0.3% compared to the DDPG method. The results verify the superiority of DF–DQN, and demonstrate the potential of the collaboration between DF and DQN.

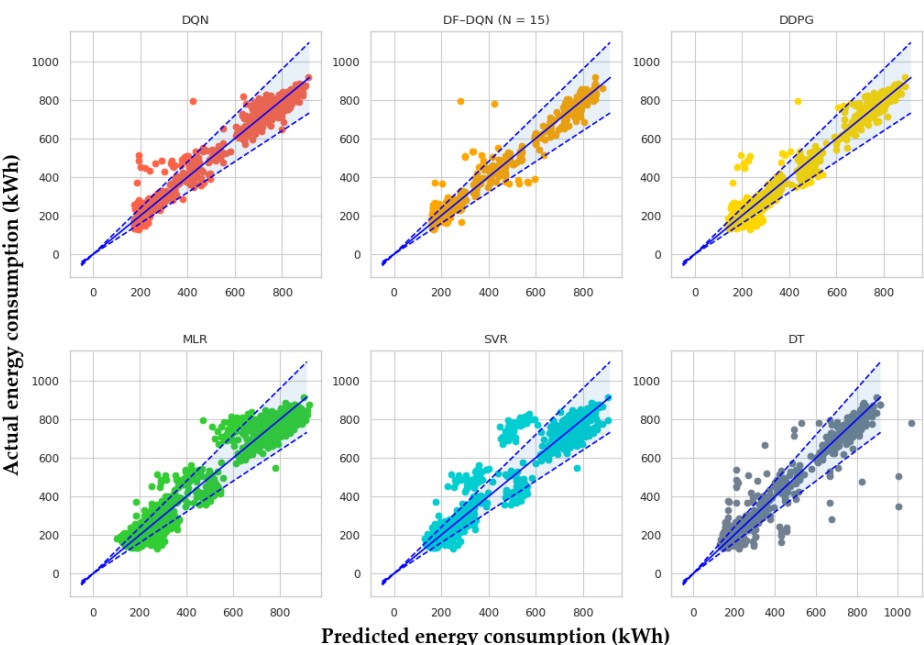

**Figure 6.** Predicted results of all methods.

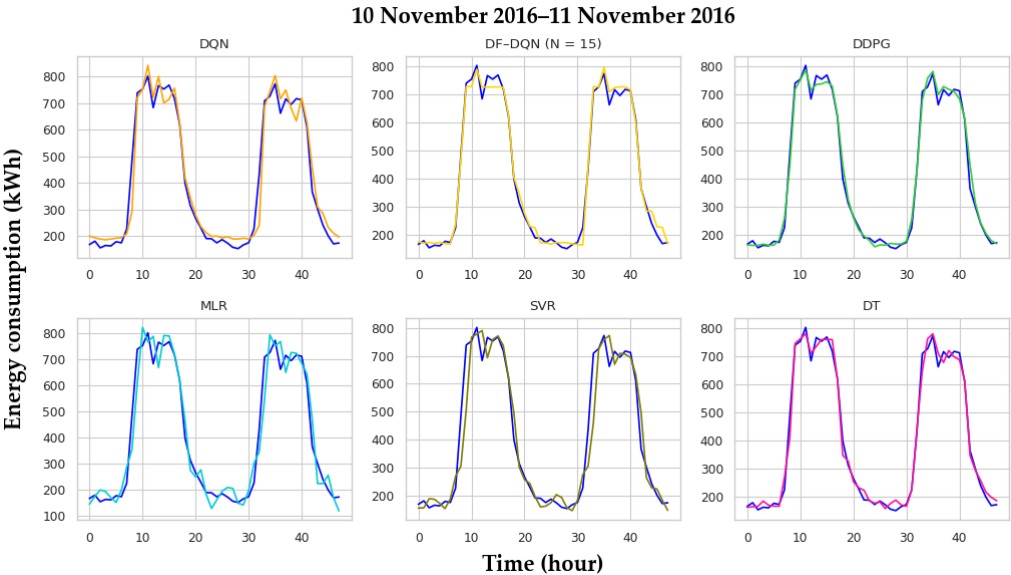

**Figure 7.** Predicted results of all methods on workdays.

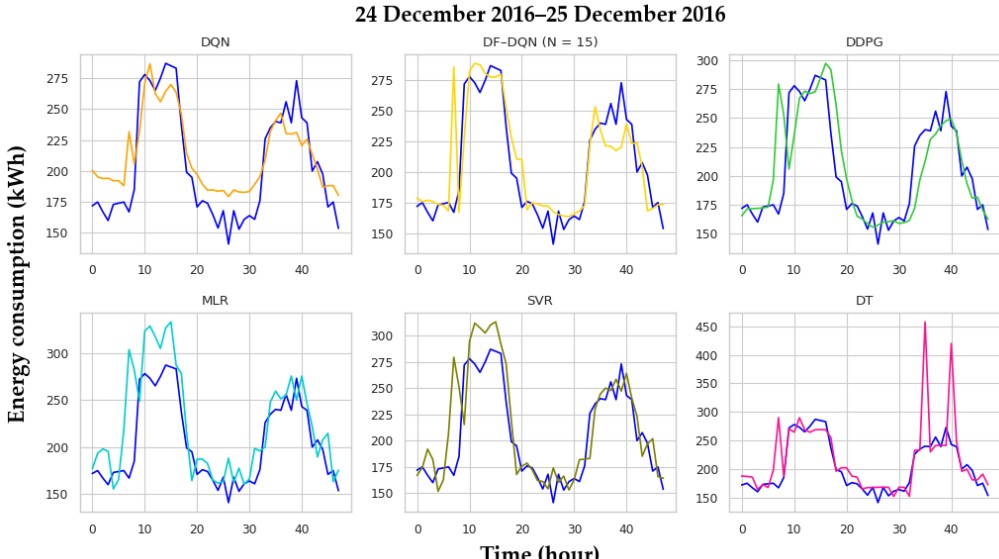

**Figure 8.** Predicted results of all methods on holidays.

**Table 6.** Prediction accuracy of all methods.

| Method | MAE | MAPE | RMSE | $R^2$ |
|---|---|---|---|---|
| MLR | 41.069 | 12.869% | 56.577 | 0.946 |
| SVR | 37.041 | 11.192% | 63.150 | 0.930 |
| DT | 26.349 | 8.868% | 50.470 | 0.959 |
| DQN | 27.942 | 9.362% | 39.869 | 0.973 |
| DDPG | 21.619 | 7.573% | 36.417 | 0.978 |
| DF–DQN ($N = 15$) | **20.432** | **7.021%** | **34.057** | **0.981** |

4.3.2. Convergence Rate and Computation Time

Figure 9 depicts MAE varying tendencies of DRL methods from the first episode, where the horizontal axis reflects iterations of the episode and the vertical axis represents MAE. It is evident that the convergence rate of the DQN method was the slowest, and the converged MAE was the highest among all DRL methods. However, the DDPG and DF–DQN methods could not be compared effectively since a very high MAE value was generated in the DDPG method. Therefore, a new figure from the fifth episode is shown to facilitate the analysis of the DDPG and DF–DQN methods. As shown in Figure 10, the convergence rate of the DDPG method was similar to that of the DF–DQN method with three state classes, and they converged near the 100th episode. The DF–DQN method with 15 state classes had the fastest convergence rate and roughly converged at the 75th episode. Further, the converged MAE value of the method was lower than other methods. It is also noteworthy that the larger the number of state classes, the faster the DF–DQN method converged, and the converged MAE value decreased as the total number of state classes increased. This is because the state classification lowered the initial value of MAE and accelerated the convergence rate of the DF–DQN method.

The convergence rates of the MLR, SVR, and DT methods are superior to DRL methods. This conclusion is presented in Table 7. The computation time of these methods is much lower than that of DRL methods. It is worth noting that the computation time of DRL methods was based on 200 episodes, as presented in Table 7. However, these DRL methods converged before the 200th episode. The computation time of the DQN method was about 625 s since the method converged at the 150th episode. Similarly, the DDPG and the DF–DQN methods with 15 state classes took 665 s and 262 s, respectively. Hence, of all DRL methods, the DF–DQN method with 15 state classes required the least computation time, and the DDPG method was slower than the DQN method. One important reason for

this is that the DF–DQN method performs in the shrunken action space, and the number of actions was significantly lower than that of the DQN method. Regarding the DDPG method, two kinds of neural networks, actor and critic networks, should be trained. These networks take long computation times, even though the total number of parameters is lower than other methods.

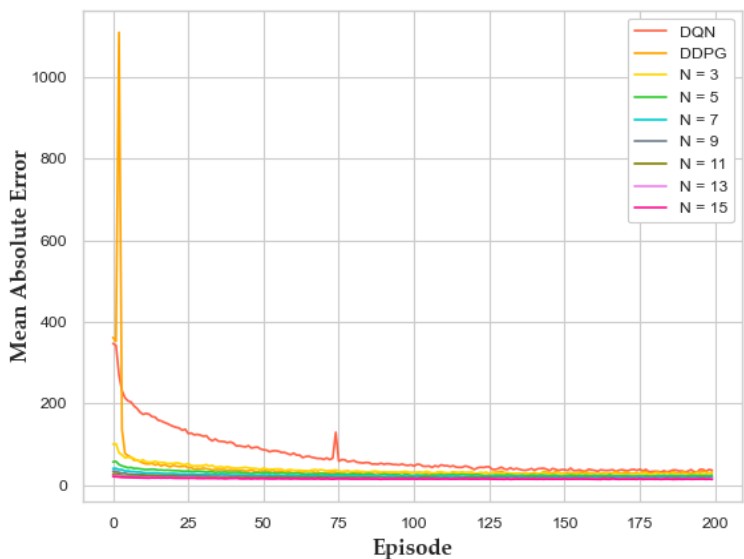

**Figure 9.** Variation tendency of MAE in DRL methods from first episode.

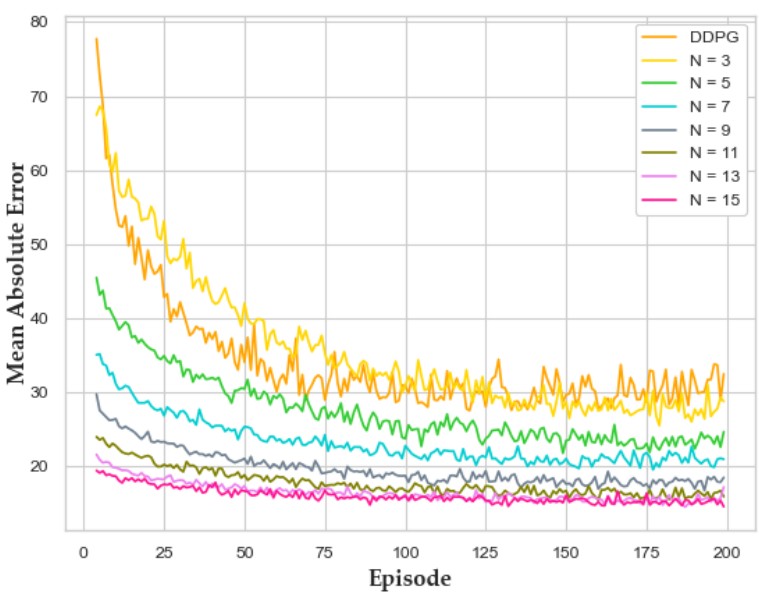

**Figure 10.** Variation tendency of MAE in DRL methods from the fifth episode.

**Table 7.** The computation time of all methods.

| Method | Computation Time |
|---|---|
| MLR | 0.07 |
| SVR | 7.734 |
| DT | 0.362 |
| DQN | 833.714 |
| DDPG | 1329.007 |
| DF–DQN (*N* = 15) | 699.529 |

## 5. Conclusions

This paper proposed a DF–DQN method to demonstrate the potential of DRL methods with discrete action space for energy consumption prediction. In the DF–DQN method, the original action space was divided into a certain number of sub-action spaces by the range of historical energy consumption data. Then, actions of the same sequence in all sub-action spaces were uniformly denoted to compose the shrunken action space. Compared with the original action space, the total number of actions in the shrunken action space was reduced greatly. Further, DF was introduced since each action has multiple meanings in the shrunken action space. State class probabilities obtained using DF can uniquely determine each action's specific meaning in the shrunken action space and map the shrunken action space to a single sub-action space, which ensures the convergence of the DF–DQN method. Moreover, the state class probabilities can also be employed to construct new states to improve the robustness of the DF–DQN method by taking into account the probabilistic process of shrinking the original action space. Based on the above operations, DF–DQN can find the optimal action quickly in the new shrunken action space.

The experimental results show that the prediction accuracy of the DF–DQN method with 15 state classes outperforms the MLR, SVR, and DT methods, even if it requires more computation time than these methods. In our experiments, for DRL methods, the DF–DQN method with 15 state classes had the highest prediction accuracy and fastest convergence rate, and required the least computation time. Specifically, compared to the DDPG method, the DF–DQN method with 15 state classes decreased MAE, MAPE, and RMSE by 5.5%, 7.3%, and 8.9%, respectively, and increased $R^2$ by 0.3%.

This study demonstrated that the DF–DQN method with discrete action space has great potential for predicting energy consumption. However, the study conducted in this paper may contain inaccuracies. The number of state classes is an additional hyperparameter and must be determined by extensive experiments. Future work will overcome the above deficiencies and explore the performance of the DF–DQN method for multi-step ahead prediction in recursive and direct multi-step manners.

**Author Contributions:** Conceptualization, K.L.; data curation, Q.F. and K.L.; formal analysis, Q.F. and K.L.; funding acquisition, J.C.; investigation, J.C., J.W., Y.L.; methodology, Q.F. and K.L.; project administration, J.W., Y.L. and Y.W.; software, Q.F. and K.L.; supervision, Q.F., J.C., J.W., Y.L. and Y.W.; validation, K.L.; writing—original draft, K.L.; writing—review & editing, Q.F. and K.L. All authors have read and agreed to the published version of the manuscript.

**Funding:** This work was financially supported by National Key R&D Program of China (No.2020YFC2 006602), National Natural Science Foundation of China (No.62072324, No.61876217, No.61876121, No.61772357), University Natural Science Foundation of Jiangsu Province (No.21KJA520005), Primary Research and Development Plan of Jiangsu Province (No.BE2020026), Natural Science Foundation of Jiangsu Province (No.BK20190942).

**Institutional Review Board Statement:** Not applicable.

**Informed Consent Statement:** Not applicable.

**Data Availability Statement:** The dataset is available at: https://github.com/gltzlike/DF-DQN-for-energy-consumption-prediction/tree/master/data (accessed on 25 January 2022).

**Conflicts of Interest:** The authors declare no conflict of interest.

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
