# Peer review of "Building Energy Consumption Prediction Using a Deep-Forest-Based DQN Method"

_buildings, doi:10.3390/buildings12020131_

Round 1

Reviewer 1 Report

This research article is based on DQN and a Novel Deep-forest-based DQN method for Building Energy Consumption Prediction. The topic is interesting to me, but the paper needs to be revised in a major way to improve the quality of the manuscript and to match the criterion of scientific publication. 

It is clear that the article type is article and not review or communication. Please clarify and list in the template line before the title.

What is the application of energy consumption prediction? I know that the goal is energy consumption is minimization, I can't find any proposed objective function for this problem. Authors should acknowledge that optimization problems have been tackled in the energy consumption domain using reinforcement learning, especially DQN :

remove the related theories sections and introduce a related work section based on energy consumption and its optimization goals. For instance, some studies utilized the energy consumption problem based on prediction.

Compare the existing related work, such as the studies listed below.

The referencing part of the paper is very poor, and a high-quality research paper published in the domain is missing.

https://www.sciencedirect.com/science/article/pii/S037877882101046X

Make a related work summary table following its merits and demerits.

The related theories sections contents should be merged throughout the manuscript, and the unnecessary should be removed as its existing work.The novelty of this paper is not clear. The difference between present work and previous Works should be highlighted.

DF is a novel decision tree ensemble method, which can be applied for classification tasks; what is the type of data you are using the DF for? Please clarify the study area and datasets with data set samples in a separate sub-section.

For instance, you can explain each step of the DQN based on the data samples using a theoretical example.

Performance analysis results should be communicated clearly using a subsection of the manuscript, For cost functions, use other performance measures metrics such as MSE, MAPE, R2 score.

The author proposed a DF-DQN method that combines DF with DQN, and introduced two novel 116

techniques in this method , How you combine the two, the mathamtics is missing. 

Clarify the state class probabilities of historical energy consumption data for energy consumption predictions. 

Make a list of the appliances and their energy consumption and comparison based on the proposed models.

described the process of data pre-processing, training, as well (line121)

as testing of DF-DQN method in detail. is this contribution. please search existing studies based on the this techniques, before claiming any contribution.

Here, the process of data pre-processing includes (line 122)

outlier detection, outlier replacement, feature extraction, and normalization ( what is new here ?)

line 124, Compared and analyzed DQN, DDPG, DF-DQN methods from three perspectives, namely prediction accuracy, convergence rate, and computation time. ( make its application and impact on energy consumption, how much energy consumption reduced etc) 

Consider the thermal comfort problem, which compares the problem in offices locations.

Line 125, then, the properties and advantages of DF-DQN method are discussed and verified.

Make a significance table comparing your approach with other scientific mechanisms used in the literature. 

Table 3 Prediction accuracy and computation time of DF-DQN method with different state classes, Accuracy of 

classification is the problem is classification; then you must use performance measures from the classification domain.

To improve the quality of the paper, authors must consider some good quality review articles, for instance, 

https://www.sciencedirect.com/science/article/pii/S136403211731362X

Title must be simple, clearer and nicer.

Spell out each acronym the first time used in the body of the paper.  

The abstract and conclusion can be rewritten to be more meaningful. The authors should add more details about their final results in the abstract. 

The abstract and conclusion should clarify what is exactly proposed (the technical contribution) and how the proposed approach is validated.

What is the motivation of the proposed work? The introduction needs to explain the main contributions of the work clearer.

 Authors must develop the framework/architecture of the proposed methods

There is a need of a flowchart and pseudocode of the proposed techniques

Research gaps, objectives of the proposed work should be clearly justified.

Lastly, 

  • English must be revised throughout the manuscript.
  • Limitations and Highlights of the proposed methods must be addressed properly.

Author Response

Response to Reviewer 1 Comments

Dear Editors and Reviewers:

Thank you for your letter and for the reviewers’ comments concerning our manuscript entitled “A Novel Deep-forest-based DQN method for Building Energy Consumption Prediction” (ID: buildings-1544145). Those comments are all valuable and very helpful for revising and improving our paper, as well as the important guidance to our research. We have studied comments carefully and have made corrections which we hope to meet with approval. Revised parts are marked in red in the manuscript. The main corrections in the paper and the responds to the reviewer’s comments are as follows:

Point 1: It is clear that the article type is article and not review or communication. Please clarify and list in the template line before the title.

Response 1: Thank you for your suggestion. We have clarified the article type in the template line before the title. (The corresponding article line number: 1)

Point 2: What is the application of energy consumption prediction? I know that the goal is energy consumption is minimization, I can't find any proposed objective function for this problem. Authors should acknowledge that optimization problems have been tackled in the energy consumption domain using reinforcement learning, especially DQN.

Response 2: Thank you for your suggestion. This paper focuses more on energy consumption prediction using deep reinforcement learning. The objective is to maximize the accumulative discount reward, which is depicted in line number 189. Besides, we acknowledge that optimization problems have been tackled using DQN. Nevertheless, many researchers also used DRL methods for energy consumption prediction and achieved satisfactory results. One limitation of current works is that many researchers only focus on DDPG method, but ignore the classical deep Q-network (DQN) method for energy consumption prediction. Therefore, we propose deep-forest-based DQN method to verify the feasibility of DQN.

Point 3: Remove the related theories sections and introduce a related work section based on energy consumption and its optimization goals. For instance, some studies utilized the energy consumption problem based on prediction.

Response 3:  Thank you for your suggestion. We have adjusted the introduction structure and introduced related works based on energy consumption and its optimization goals in 1.1.2 (The corresponding article line number: 123-160). Meanwhile, after our full discussion and thinking twice, we believe the related theories sections can help researchers better understand the basic principles of DF and DQN, then DF-DQN method can be more acceptable. Therefore, we suggest retaining the related theories.

Point 4: Compare the existing related work, such as the studies listed below. The referencing part of the paper is very poor, and a high-quality research paper published in the domain is missing. https://www.sciencedirect.com/science/article/pii/S037877882101046X

Response 4: Thank you for your suggestion. We have added the high-quality research paper in 1.1.2. Meanwhile, we also have appended many extra related works in the introduction. (The corresponding article line number: 136)

Point 5: Make a related work summary table following its merits and demerits.

Response 5: Thank you for your suggestion. We have made the related work summary table in the introduction. (The corresponding article line number: 121)

Point 6: The novelty of this paper is not clear. The difference between present work and previous Works should be highlighted.

Response 6: Thank you for your suggestion. We have emphasized the innovation of our method in 1.1.3 (The corresponding article line number: 171-178). Meanwhile, the difference between present work and previous Works is highlighted in the introduction. (The corresponding article line number: 117-120, 157-160) 

Point 7: DF is a novel decision tree ensemble method, which can be applied for classification tasks; what is the type of data you are using the DF for? Please clarify the study area and datasets with data set samples in a separate sub-section.

Response 7: Thank you for your suggestion. We have added the 3.3.2 section to clarify DF classifier. (The corresponding article line number: 374)

Point 8: Performance analysis results should be communicated clearly using a subsection of the manuscript, For cost functions, use other performance measures metrics such as MSE, MAPE, R2 score.

Response 8: Thank you for your suggestion. We have analyzed predicted results in 4.3.1 and added descriptions of the results of DF-DQN method (The corresponding article line number: 471-498). Moreover, We have added MAPE, R2 in our experiments. Since MSE has a similar meaning as RMSE, we do not add it. (The corresponding article line number: 462)

Point 9: The author proposed a DF-DQN method that combines DF with DQN, and introduced two novel techniques in this method, How you combine the two, the mathamtics is missing.

Response 9: Thank you for your suggestion. The combination of DF and DQN is mainly a process instead of mathematics. The process is described in section 3.3.2. It is worth noting that deep forest is employed to shrink the original action space and ensure the convergence of the DF-DQN method. (The corresponding article line number: 382)

Point 10: Clarify the state class probabilities of historical energy consumption data for energy consumption predictions.

Response 10: Thank you for your suggestion. We have clarified the purposes of state class probabilities in section 3.3.2. (The corresponding article line number: 395)

Point 11: Make a list of the appliances and their energy consumption and comparison based on the proposed models.

Response 11: Thank you for your suggestion. Our current research does not involve specific appliances. In this study, we are more concerned with the energy consumption prediction for individual building. After our full discussion, we do not add the energy consumption and comparison of the appliances in the paper. In the future control work, we will analyze and experiment on this problem.

Point 12: Described the process of data pre-processing, training, as well (line121) as testing of DF-DQN method in detail. is this contribution. please search existing studies based on the this techniques, before claiming any contribution.

Here, the process of data pre-processing includes (line 122) outlier detection, outlier replacement, feature extraction, and normalization (what is new here ?)

Line 124, Compared and analyzed DQN, DDPG, DF-DQN methods from three perspectives, namely prediction accuracy, convergence rate, and computation time. (make its application and impact on energy consumption, how much energy consumption reduced etc)

Line 125, then, the properties and advantages of DF-DQN method are discussed and verified. Make a significance table comparing your approach with other scientific mechanisms used in the literature. 

Response 12: Thank you for your suggestion. After careful consideration, we have rewritten this part to show the details of DF-DQN method. On the one hand, we have found a similar data pre-processing procedure in other literature. On the other hand, we consider the experimental results to be facts rather than contributions (The corresponding article line number: 171-178). Besides, the results of comparison between DF-DQN method and other scientific mechanisms are detailed in Table 6 and Table 7. (The corresponding article line number: 531)

Point 13: Consider the thermal comfort problem, which compares the problem in offices locations.

Response 13: We are currently focusing on energy consumption prediction for individual building. After our full discussion and thinking twice, we believe that thermal comfort problem is more appropriate for future control work.

Point 14: Table 3 Prediction accuracy and computation time of DF-DQN method with different state classes, Accuracy of classification is the problem is classification; then you must use performance measures from the classification domain.

Response 14: Thank you for your suggestion. From the perspective of classification, the higher the classification accuracy, the better. However, the final results of DF-DQN are not determined by the classification accuracy. For instance, the DF-DQN with 15 state classes has the highest prediction accuracy, although its classification accuracy is 76.462%. The reason is that as the number of classes increases, the classification accuracy decreases, but at the same time, the original action space shrinks more. So, there is a balance between the classification accuracy and the shrink of action space. After careful consideration, we believe that the details of the DF classification results are unnecessary.

Point 15: To improve the quality of the paper, authors must consider some good quality review articles, for instance, https://www.sciencedirect.com/science/article/pii/S136403211731362X

Response 15: Thank you for your suggestion. We have introduced this review article in our paper. (The corresponding article line number: 67)

Point 16: Title must be simple, clearer and nicer.

Response 16: Thank you for your suggestion. We have decided to change the title of our paper to “Building Energy Consumption Prediction using Deep-forest-based DQN method”.

Point 17: Spell out each acronym the first time used in the body of the paper.  

Response 17: Thank you for your suggestion. We have checked and spelled out each acronym the first time used in our paper.

Point 18: The abstract and conclusion can be rewritten to be more meaningful. The authors should add more details about their final results in the abstract. The abstract and conclusion should clarify what is exactly proposed (the technical contribution) and how the proposed approach is validated.

Response 18: Thank you for your suggestion. We have rewritten the abstract and conclusion to emphasize our technical contributions. Moreover, extra measures metrics are added to this paper to verify the performance of DF-DQN method, and detailed experimental results are described in the abstract and conclusion. (The corresponding article line number: 15-28, 567-593)

Point 19: What is the motivation of the proposed work? The introduction needs to explain the main contributions of the work clearer.

Response 19: Thank you for your suggestion. We have added descriptions of motivation

in the introduction (The corresponding article line number: 117-120, 157-160, 168-170). Moreover, we have added more clarity to our contributions in the introduction (The corresponding article line number: 172-180).

Point 20: Authors must develop the framework/architecture of the proposed methods. There is a need of a flowchart and pseudocode of the proposed techniques.

Response 20: Thank you for your suggestion. We have added a flowchart to clarify DF-DQN method. (The corresponding article line number: 390). And the pseudocode is exhibited in 3.4 section. (The corresponding article line number: 431)

Point 21: Research gaps, objectives of the proposed work should be clearly justified.  

Response 21: Thank you for your suggestion. We have added descriptions of research gaps and objectives in the introduction (The corresponding article line number: 117-120, 157-160, 168-170)

Point 22: English must be revised throughout the manuscript.

Response 22: Thank you for your suggestion. This paper has been checked by our native English-speaking colleague.

Point 23: Limitations and Highlights of the proposed methods must be addressed properly.

Response 23: Thank you for your suggestion. We have rewritten the abstract and conclusion to analyze our method. Highlights and limitations are described in the abstract and conclusion. (The corresponding article line number: 15-28, 567-593).

Reviewer 2 Report

In this paper, authors proposed a deep-forest-based DQN (DF-DQN) method, which can obtain higher prediction accuracy than DRL methods with continuous action space, and take less computation time. In DF-DQN, two novel techniques are introduced.

Firstly, the original action space is replaced with the shrunken action space, which means that fewer parameters need to be trained, and so the computation time is reduced. Secondly, class probabilities of original states, which are obtained by deep forest (DF), are employed to construct new states for further energy consumption prediction.

2. Please ensure that all variables/symbols introduced in the manuscript are properly explained and the index of each symbol is correct and consistent in order to avoid confusion.

3. Please include information regarding the APIs, libraries and packages used for this project with their respective version numbers. Additionally, any details regarding the hardware used for those experiments are welcome. Information that could help reproduce and further understand this study on a programming level would be beneficial for future research and experiments.

4. In this manuscript, the authors proposed a deep-forest-based DQN (DF-DQN) method to obtain higher prediction accuracy than DRL methods with continuous action space, and take less computation time. The manuscript is interesting, however, some amendments are required. The related work section should be improved by better discussing the limitations of the existing methods and discussing more related works.

Several references about hybrid deep learning usage and Energy Consumption Prediction model can be added [1][2].

[1] Ozcan, A.; Catal, C.; Kasif, A. Energy Load Forecasting Using a Dual-Stage Attention-Based Recurrent Neural Network.
 Sensors 2021, 21, 7115. https://doi.org/10.3390/s21217115

[2] Ozcan, A., Catal, C., Donmez, E. et al. A hybrid DNN–LSTM model for detecting phishing URLs.  Neural Comput & Applic (2021). https://doi.org/10.1007/s00521-021-06401-z

5. It is not clear why the authors selected this deep-forest-based DQN (DF-DQN) method architecture, why not other architectures. The authors should justify it by discussing some references or experimentally

6. The experimental analysis section should be strengthened. The authors can evaluate different architectures of energy consumption prediction

Author Response

Response to Reviewer 2 Comments

Dear Editors and Reviewers:

Thank you for your letter and for the reviewers’ comments concerning our manuscript entitled “A Novel Deep-forest-based DQN method for Building Energy Consumption Prediction” (ID: buildings-1544145). Those comments are all valuable and very helpful for revising and improving our paper, as well as the important guidance to our research. We have studied comments carefully and have made corrections which we hope to meet with approval. Revised parts are marked in red in the manuscript. The main corrections in the paper and the responds to the reviewer’s comments are as follows:

Point 1: Please ensure that all variables/symbols introduced in the manuscript are properly explained and the index of each symbol is correct and consistent in order to avoid confusion.

Response 1: Thank you for your suggestion. We have checked all variables/symbols introduced in our manuscript.

Point 2: Please include information regarding the APIs, libraries and packages used for this project with their respective version numbers. Additionally, any details regarding the hardware used for those experiments are welcome. Information that could help reproduce and further understand this study on a programming level would be beneficial for future research and experiments.

Response 2: Thank you for your suggestion. We have added the hardware platform and package version in 4.1 section to be beneficial for future research and experiments. (The corresponding article line number: 450)

Point 3: In this manuscript, the authors proposed a deep-forest-based DQN (DF-DQN) method to obtain higher prediction accuracy than DRL methods with continuous action space, and take less computation time. The manuscript is interesting, however, some amendments are required. The related work section should be improved by better discussing the limitations of the existing methods and discussing more related works. Several references about hybrid deep learning usage and Energy Consumption Prediction model can be added [1][2].

[1] Ozcan, A.; Catal, C.; Kasif, A. Energy Load Forecasting Using a Dual-Stage Attention-Based Recurrent Neural Network.Sensors 2021, 21, 7115. https://doi.org/10.3390/s21217115

[2] Ozcan, A., Catal, C., Donmez, E. et al. A hybrid DNN–LSTM model for detecting phishing URLs.  Neural Comput & Applic (2021). https://doi.org/10.1007/s00521-021-06401-z

Response 3: Thank you for your suggestion. We have adjusted the introduction structure and introduced these papers. Meanwhile, we also have appended many extra related works in our introduction. (The corresponding article line number: 56-79, 123-160)

Point 4: It is not clear why the authors selected this deep-forest-based DQN (DF-DQN) method architecture, why not other architectures. The authors should justify it by discussing some references or experimentally

Response 4: Thank you for your suggestion. We have added explanations in our paper. (The corresponding article line number: 375)

Point 5: The experimental analysis section should be strengthened. The authors can evaluate different architectures of energy consumption prediction.

Response 5: Thank you for your suggestion. We have added measured metrics, such as MAPE and R2 score, to our experiments. Besides, multiple linear regression (MLR), support vector regression (SVR), and decision tree (DT) have been added to the experiments. (The corresponding article line number: 531)

Reviewer 3 Report

The article is written on current topics, the methodology and research methods have been analyzed sufficiently. The sources used in the article reflect the research problem. In general, the article is written at a high scientific level. The Conclusions should more clearly describe the results obtained and indicate their scientific novelty, with justification, how they differ from existing ones, and what they allow to achieve.

Author Response

Response to Reviewer 3 Comments

Dear Editors and Reviewers:

Thank you for your letter and for the reviewers’ comments concerning our manuscript entitled “A Novel Deep-forest-based DQN method for Building Energy Consumption Prediction” (ID: buildings-1544145). Those comments are all valuable and very helpful for revising and improving our paper, as well as the important guidance to our research. We have studied comments carefully and have made corrections which we hope to meet with approval. Revised parts are marked in red in the manuscript. The main corrections in the paper and the responds to the reviewer’s comments are as follows:

Point 1: The Conclusions should more clearly describe the results obtained and indicate their scientific novelty, with justification, how they differ from existing ones, and what they allow to achieve

Response 1: Thank you for your suggestion. We have rewritten the conclusions to represent the novelty of DF-DQN method. Meanwhile, we have added measures metrics in experiments, and the detailed experimental results are described in the conclusions. (The corresponding article line number: 567-593)

Round 2

Reviewer 1 Report

The authors have addressed all my concerns, and I recommend the manuscript for publication.

Regards and Best wishes